# Multi-scale model suggests the trade-off between protein and ATP demand as a driver of metabolic changes during yeast replicative ageing

Barbara Schnitzer[1,2]☯, Linnea Österberg[1,2,3]☯, Iro Skopa[1,2], Marija Cvijovic[1,2]*

**1** Department of Mathematical Sciences, Chalmers University of Technology, Gothenburg, Sweden, **2** Department of Mathematical Sciences, University of Gothenburg, Gothenburg, Sweden, **3** Department of Biology and Biological Engineering, Chalmers University of Technology, Gothenburg, Sweden

☯ These authors contributed equally to this work.
* marija.cvijovic@chalmers.se

**Data Availability Statement:** Computational code is available at: https://github.com/cvijoviclab/IntegratedModelMetabolismAgeing All data are included in the manuscript.

## Abstract

The accumulation of protein damage is one of the major drivers of replicative ageing, describing a cell's reduced ability to reproduce over time even under optimal conditions. Reactive oxygen and nitrogen species are precursors of protein damage and therefore tightly linked to ageing. At the same time, they are an inevitable by-product of the cell's metabolism. Cells are able to sense high levels of reactive oxygen and nitrogen species and can subsequently adapt their metabolism through gene regulation to slow down damage accumulation. However, the older or damaged a cell is the less flexibility it has to allocate enzymes across the metabolic network, forcing further adaptions in the metabolism. To investigate changes in the metabolism during replicative ageing, we developed an multi-scale mathematical model using budding yeast as a model organism. The model consists of three interconnected modules: a Boolean model of the signalling network, an enzyme-constrained flux balance model of the central carbon metabolism and a dynamic model of growth and protein damage accumulation with discrete cell divisions. The model can explain known features of replicative ageing, like average lifespan and increase in generation time during successive division, in yeast wildtype cells by a decreasing pool of functional enzymes and an increasing energy demand for maintenance. We further used the model to identify three consecutive metabolic phases, that a cell can undergo during its life, and their influence on the replicative potential, and proposed an intervention span for lifespan control.

## Author summary

Understanding the complex and multi-scale nature of ageing requires the integration of key biological process and ageing hallmarks occurring on different scales. Hallmarks of ageing are conserved across all kingdoms of life and to elucidate the mechanisms of ageing we turn to unicellular organisms like yeast *Saccharomyces cerevisiae*. In this work, we

**Funding:** This work was supported by the Swedish Research Council (VR2016-03744 and VR2017-05117) to MC and the Swedish Foundation for Strategic Research (FFL15-0238) to MC. The funders had no role in study design, data collection and analysis, decision to publish, or preparation of the manuscript.

**Competing interests:** The authors have declared that no competing interests exist.

present a comprehensive, multi-scale model of yeast replicative ageing integrating metabolism, nutrient signalling and damage accumulation. The proposed model demonstrates the relevance of the regulatory layer for achieving average lifespans and generation times of yeast-wild type cells. Further, we show that metabolic phases are tightly linked to the replicative lifespan and we propose that enzyme perturbation in the specific phase can result in prolonged or shorten lifespan.

## 1 Introduction

Cellular ageing is a complex multifactorial process affected by an intertwined network of effectors such as protein translation, protein quality control, mitochondrial dysfunction, and metabolism. Due to the conserved nature of hallmarks of ageing [1] unicellular organisms, such as the yeast *Saccharomyces cerevisiae*, have served as model organisms to gain deeper understanding of their synergistic effects and consequently mechanisms of ageing on a cellular level [2–6].

Loss of proteostasis is recognised as one of the hallmarks of replicative ageing [1, 7, 8], and is linked to the accumulation of damaged proteins over time [9, 10]. In yeast, a driving mechanism for the growing damage burden is the asymmetric distribution of damaged components between mother and daughter cell [10, 11]. An important precursor of protein damage is oxidative stress that is shown to increase with age [12–14] and is a byproduct of metabolic activity in the cells' mitochondria [14–17]. Reactive oxygen and nitrogen species (ROS/RNS) are one of the most well-studied byproducts to which cells are constantly exposed even under normal conditions [18, 19]. The ability of cells to maintain protein homeostasis in response to intrinsic cellular and environmental factors, which accumulate over time, is one of the main determinants of lifespan [20]. Nutrient-sensing pathways are main contributors to the maintenance of the proteome during ageing. When inactivated, they affect a multitude of downstream processes, resulting in the cellular loss of proteostasis. Thus, they are one of the earliest events dictating ageing progression. To combat the loss of proteostasis associated with cellular ageing, cells have multiple stress-responsive mechanisms. For instance, Msn2 and Msn4, as general stress response proteins, as well as Yap1 and Skn7, as specific oxidative stress response proteins, are able to react to high levels of oxidative stress and can enhance the removal of ROS/RNS via adaption of gene expression [21, 22]. Nevertheless, the accumulation of protein damage during replicative ageing cannot be prevented and in turn, affects the metabolism and its activity. It has been shown that cells to undergo distinct metabolic phases during their replicative life [23, 24], exhibiting a switch from energy production via fermentation to cellular respiration.

Hitherto, many mathematical models describing protein damage accumulation (reviewed in [25, 26]), signalling pathways (reviewed in [27, 28]) and metabolism [29] have been developed. Flux balance analysis (FBA) models have been extensively applied to predict fluxes through genome-scale reconstructions of metabolic networks of many different organisms and conditions [29–32]. In order to improve their predictive power, they have been extended by additional constraints, such as enzymatic and regulatory constraints [33–37] and lately proteome constraints [38]. Extensive and condition-specific regulatory constraints can be obtained by reconstructions of signalling pathways. Due to their size and availability of vast qualitative data, they are typically represented and simulated using logic or Boolean modelling [39–42]. Recent studies aimed at combining those two approaches into so-called hybrid modes, to understand the connection between signalling and the metabolism [37, 43].

However, most existing models are answering isolated questions regarding the cellular metabolism and its regulation with focus on short time scales compared to the lifespan of a cell. Further, they have not been used to understand damage accumulation over long time scales, i.e. ageing, despite the tight connection between the metabolism, ROS/RNS and damage accumulation. Recently, an FBA-based model was used to rationalise metabolic data at distinct time points during the replicative life of yeast cells [23]. While the study is based on the qualitative interpretation of the acquired data, mechanistic insights and dynamics are missing. On the other hand, dynamic models of protein damage accumulation have been applied to investigate replicative ageing on a single-cell level and the effect on the population level, however disregarding metabolic effects [44–48].

Taken together, while existing models have greatly improved our understanding of these key processes, they have also revealed gaps in the understanding of the complex interactions between them, as they are mainly studied individually and, in addition, lack both the complexity of the dynamics and the effects of the crosstalk between multiple components.

To overcome this gap, and to study the complex interplay and feedback between the metabolism and replicative ageing, in the context of damage accumulation and reactive oxygen species, we built a multi-scale model of yeast replicative ageing, that includes an enzyme-constrained FBA model, a Boolean model of nutrient signalling pathways and dynamic model of protein damage accumulation and cell growth. The model can simulate the lifespan of a cell being controlled by the metabolism, allowing to explore metabolic changes as the cell ages and becomes exposed to oxidative stress and protein damage.

## 2 Results

### Construction of a multi-scale yeast replicative ageing model

To elucidate how nutrient and stress signalling, metabolism and protein damage accumulation influence and regulate each other during the life of a cell, we developed a multi-scale model of yeast replicative ageing (yMSA). The model consists of three interconnected modules: a Boolean model of the signalling network, an enzyme-constrained flux balance model of the central carbon metabolism and a dynamic model of growth and protein damage accumulation with discrete cell divisions (Fig 1 and Table 1).

The first step in the construction was to extend a hybrid model of the central carbon metabolism and nutrient signalling in budding yeast [37] by reactions and components attributed to the production and removal of ROS and RNS, as well as the signalling response to it. In particular, we built a Boolean model of the Yap1 and Sln1 pathways (Fig 1A) and incorporated it in the already existing Boolean model of the nutrient sensing pathways PKA, Snf1 and TOR [37]. While Msn2 and Msn4 are part of the PKA pathway, we also included the crosstalk to the Yap1 pathway. In addition, we summarised reactions that create and remove ROS/RNS (Fig 1B) and added them to the existing model of the central carbon metabolism [37]. The modules of signalling and metabolism are connected by a transcriptional layer, that modifies the enzyme consumption in the metabolic model depending on the binary activities of transcription factors in the Boolean model by imposing regulatory constraints. In turn, the optimal fluxes of the enzyme constrained FBA (ecFBA) model determine the states of input components in the Boolean model.

To validate the extensions, we demonstrated that the steady-state activity of the transcription factors in the Boolean model is consistent with the presence or absence of oxidative stress (S1 Fig). In addition, we confirmed that the included production, sensing and removal of oxidative stress in the metabolic and signalling network does not affect the exchange fluxes

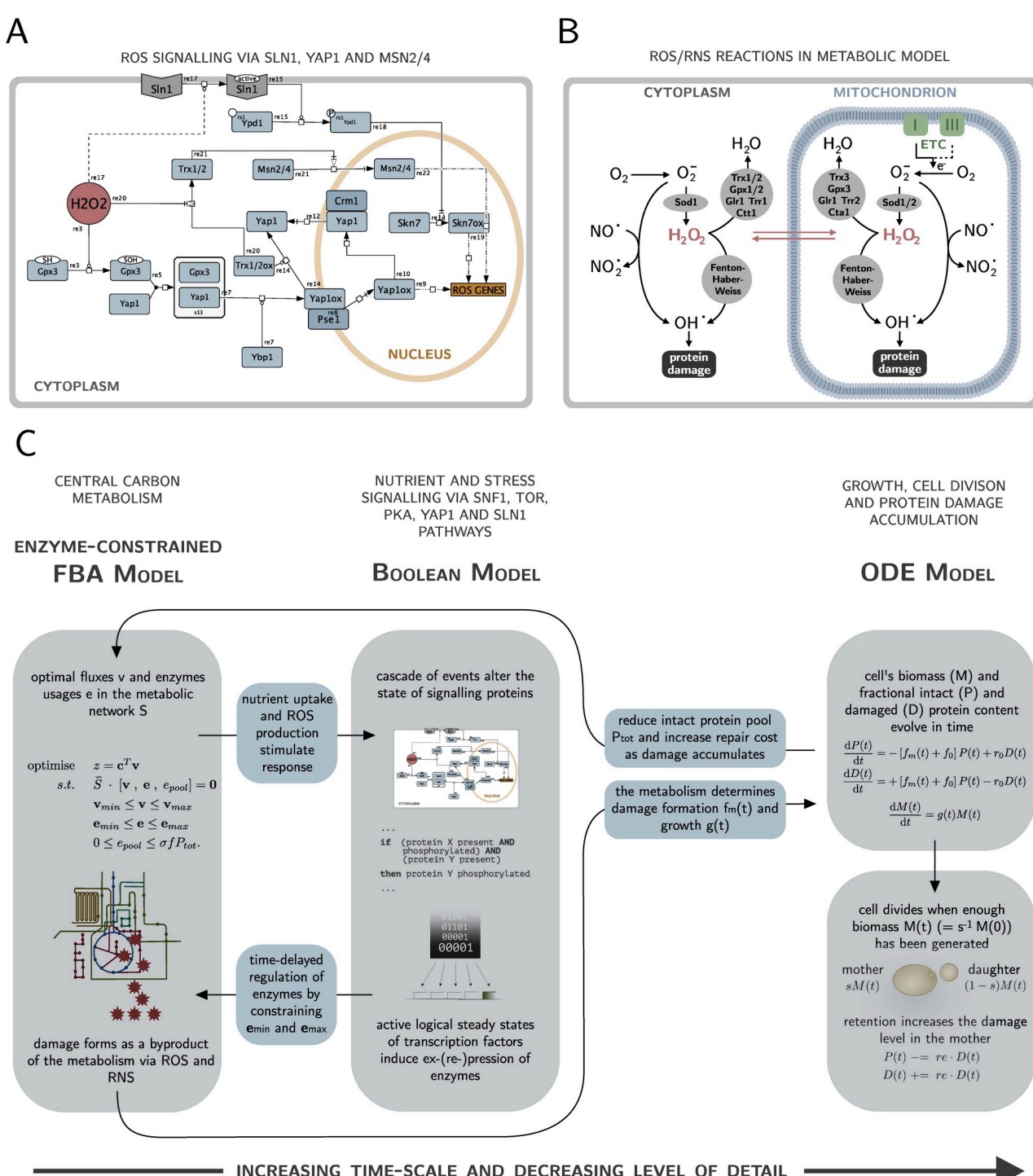

**Fig 1. Multi-scale model construction.** (A) Yap1 and Sln1 signalling in response to oxidative stress via $H_2O_2$. The two pathways were added to Boolean signalling network. Trx1/2 exhibits cross-talk to Msn2/4, a component that is also part of the nutrient sensing pathway PKA. The figure is made with Cell Designer [72]. (B) ROS/RNS reactions that were added to the enzyme-constrained FBA model. The cell is exposed to oxidative stress as a consequence of electron leakage in the electron transport chain (ETC). (C) Schematic view of one time step in the multi-scale model. The enzyme-constrained FBA fluxes based on the current fraction of intact and damaged proteins determine the input states of the Boolean signalling layer. A set of Boolean rules alter the states of the signalling proteins, that eventually induce gene ex-/repression via a transcriptional layer, leading to constraints in individual enzyme usages. Solving the regulated enzyme-constrained FBA gives rise to a growth rate as well as a metabolic damage formation rate, that feed into the ODE model of growth and damage accumulation that is then solved for one time step. If the cell has accumulated enough biomass, the cell divides in an instantaneous event. Iterating the model over time-steps until the model becomes infeasible corresponds to a lifespan simulation.

**Table 1. The size of the yeast multi-scale model of ageing (yMSA) and how it is composed of three modules.**

| Yeast multi-scale model of ageing (yMSA) | | | |
|---|---|---|---|
| Type | Total number | Composition and Description | Module |
| **Components** | 226 | 86 signalling proteins and input metabolites | Boolean |
| | | 137 metabolites and pseudo metabolites | ecFBA |
| | | 3 states: biomass, intact and damaged protein fraction | ODE |
| **Enzymes** | 141 | 140 catalysing enzymes and the enzyme pool | ecFBA |
| **Rules** | 122 | interactions between signalling proteins | Boolean |
| **Fluxes** | 375 | chemical reactions and pseudo reactions | ecFBA |

measured in a chemostat experiment [73], a widely used experiment to validate metabolic models (S2 Fig).

In the second step, we connected the described regulated ecFBA model to an ordinary differential equations (ODE) model of cell growth and protein damage accumulation with discrete cell divisions (Fig 1C). Here, the parameters of the ODE model depend on the optimal fluxes through the regulated ecFBA model. After solving the ODE for one time step given those parameters, the resulting fraction of intact and damaged proteins constrain the regulated ecFBA model for the next time step.

To simulate the whole lifespan of a cell, the model was iterated over time steps. Cell death automatically occurs when the ecFBA becomes infeasible, caused by a too high protein damage burden such that the cell is not able to generate enough energy for maintenance and growth anymore. Each time step is based on the assumption that the signalling and metabolic adaptions happen on a much faster time scale than damage accumulation and ageing. We furthermore accounted for a delay between an actual signalling event and its effect on the metabolism through gene expression by applying the regulation step only after $n_{delay}$ time steps.

All mathematical and computational details of each individual module as well as the crucial interfaces can be found in the Methods section and in S1 and S2 Texts.

## The model predicts features of replicative ageing with distinct metabolic phases

To validate our multi-scale model, we tested if it can reproduce features of replicative ageing in yeast cells. We focused firstly on the number of divisions (replicative lifespan) and the time between divisions (generation time), and secondly on metabolic paths cells use to gain energy in the course of damage accumulation.

In each simulation, we started with a damage-free cell and let the model evolve over time until cell death occurs. The objective of the metabolic model is always maximal growth. Given the signalling and metabolic networks, we tested the effect of the ODE model parameters (non-metabolic damage formation $f_0$ and damage repair $r_0$) on the lifespan. Our model predicts replicative lifespans between 17–32 cell divisions in the tested parameter regime (Fig 2A), in accordance with measured yeast wildtype lifespans of on average around 23 divisions [74–76]. The slower damage forms, the higher is the replicative potential of the cell. An increase in the repair rate has a positive effect on the lifespan, however it cannot counteract the large increase of dysfunctional proteins in mother cells caused by the asymmetric damage segregation at cell division, being a major driver of replicative ageing [10, 11, 77, 78]. Only if repair rates are more than approximately one order of magnitude higher than non-metabolic formation rates the damage burden of retention can be overcome (S3D Fig).

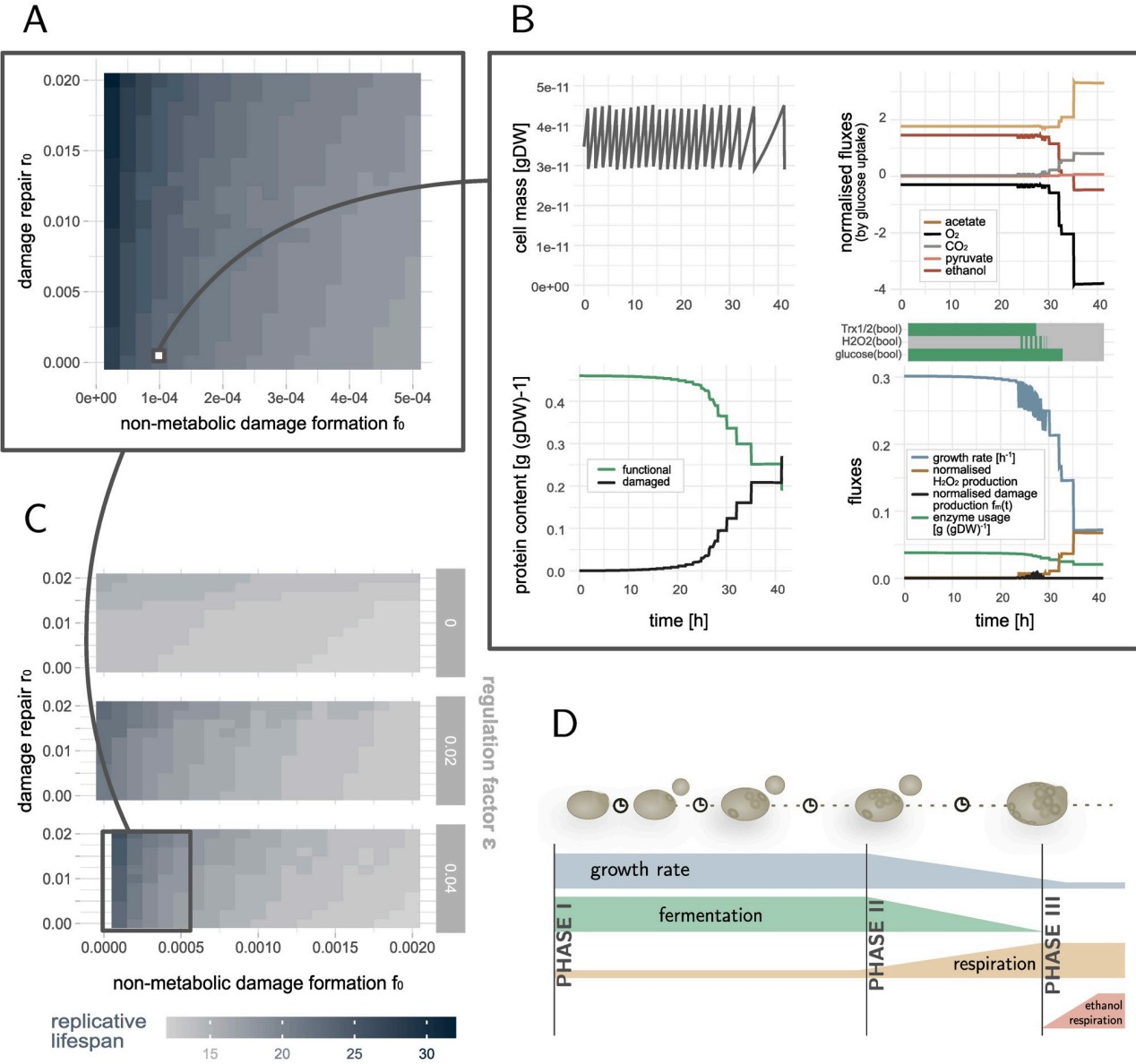

**Fig 2. Lifespan simulations of yeast wildtype cells.** (A) Replicative lifespans for typical yeast wildtype cells with varying damage repair $r_0$ and non-metabolic damage formation rates $f_0$. The model reproduces 17–32 divisions. (B) Zoom into variables of all three model parts over time for a specific exemplary parameter set ($f_0 = 0.0001$, $r_0 = 0.0005$): cell mass, fraction of intact and damaged proteins, growth rate, exchange fluxes normalised by the glucose uptake rate in the metabolic model ($> 0$: production rates, $< 0$: uptake rates), functional enzyme pool and input signals received by the signalling network (green: present, grey: not present). As cells age they accumulate damage, the growth rate drops and the metabolism needs to adapt. (C) Zoom out for varying damage repair $r_0$ and non-metabolic damage formation rates $f_0$ and regulation factors $\epsilon$. If the tile is not filled ($f_0 = 0.0$ and $\epsilon = 0.04$), the simulated cell did not stop dividing in the simulation time. Stronger regulation increases the replicative lifespan and wildtype cells with more than 22 divisions cannot be achieved in this resolution if the regulation factor $\epsilon < 0.04$. (D) Schematic view of the metabolic phases a cells undergoes during its replicative life: from maximal growth and fermentation (I) it slowly switches to respiration when the growth rate drops (II) until it eventually can also take up ethanol to produce energy close to cell death (III).

To illustrate the dynamics of the model's components, we selected a representative wildtype cell with 23 divisions and an average generation time of around 1.5 hours and followed the optimal fluxes through the metabolic network, the signals that the cell senses and its protein composition over time (Fig 2B). A typical *in silico* cell starts with a fully functional protein

pool that mediates chemical reactions in the metabolic network. The resources can be fully exploited and allow for high growth and cell division rates. Over time, as damage accumulates, the functional pool shrinks continuously, passing a point when the cell needs to decrease the glucose uptake and growth rates, and metabolic fluxes have to be redistributed. Along with an increasing demand of ATP for repairing damage, the cell needs to become more efficient in the ATP production. While most energy during the maximal growth phase is produced via fermentation (high production of ethanol), respiration gets more and more prominent when the growth rate drops (increase in oxygen $O_2$ uptake and acetate and carbon dioxide $CO_2$ production). Consequently, damage is increasingly produced by ROS/RNS and the cell signals oxidative stress, followed by an increased use of enzymes that are needed to remove those again. Older cells with low growth rates produce less damage, and stress signalling is not active anymore. Instead, cells take in less nutrients and eventually signal glucose limiting conditions. That old cells show signatures of starved cells was recently confirmed experimentally [79]. Close to death, cells also take in ethanol to produce energy and prolong lifespan. However, they can only grow slowly (Fig 2B).

Decreasing growth rates in our model induce slower generation times, i.e. times between cell divisions. In particular, the last few divisions before cell death last significantly longer, as observed previously [75, 80]. In contrast to published models of protein damage accumulation that have to assume this decline in growth [46, 47], here it is a direct output of the model.

Taken together, our model can reproduce characteristics of replicative ageing in wildtype yeast cells, being a consequence of a shrinking pool of functional proteins available for the metabolism and an increasing demand of energy for non-growth associated maintenance such as damage repair. In particular, simulated cells undergo distinct metabolic phases: (I) maximal growth phase mainly mediated by fermentation, (II) switching phase to respiration characterised by a mixed metabolism, a decrease in growth rate and an increase in ROS/RNS production, and (III) slow growth phase defined by ethanol uptake (Fig 2D).

## Metabolic regulation by the signalling network is beneficial for the replicative lifespan

To identify the effect of stress signalling on the replicative lifespan, we simulated cells with varying regulation strengths. By regulation strength we mean the magnitude of the constraints on the protein abundance caused by stress signalling affecting the metabolic model. The regulation strength is different for every protein depending on the solution space, and is controlled by a global regulation factor $\epsilon$, as specified in Eq (1). The model showed that increasing the strength of the regulation of enzymes is beneficial for the replicative lifespan up to $\epsilon \approx 0.04$ (Figs 2C and 3A), corresponding to constraining a down- or upregulated enzyme in its usage from above or below respectively by 4% of its enzyme variability. Wildtype cells, i.e. cell with around 23 divisions and an average generation time of 1.5 to 2h, can only be generated with this maximal regulation strength and low damage formation rates (Fig 2C and S3B Fig). We observed that regulation has a particularly strong effect on the maximal growth phase, i.e. phase I. Due to regulation the amount of damage that is produced in this phase is reduced while the amount of divisions increased (Fig 3A and S3C Fig). An increase in the number of divisions in phase II is only possible for a large impact of regulation on the metabolism ($\epsilon > 0.02$). There is a similar maximal amount of damage that a cell can tolerate (damage at end of phase III) regardless of the regulation, indicating that a decreased damage accumulation in the early life of the cell is essential for the replicative lifespan.

The model cannot handle higher regulation factors than $\epsilon = 0.05$. If enzymes are too heavily constrained, i.e. $\epsilon$ is large, the model sooner or later becomes over-constrained and infeasible

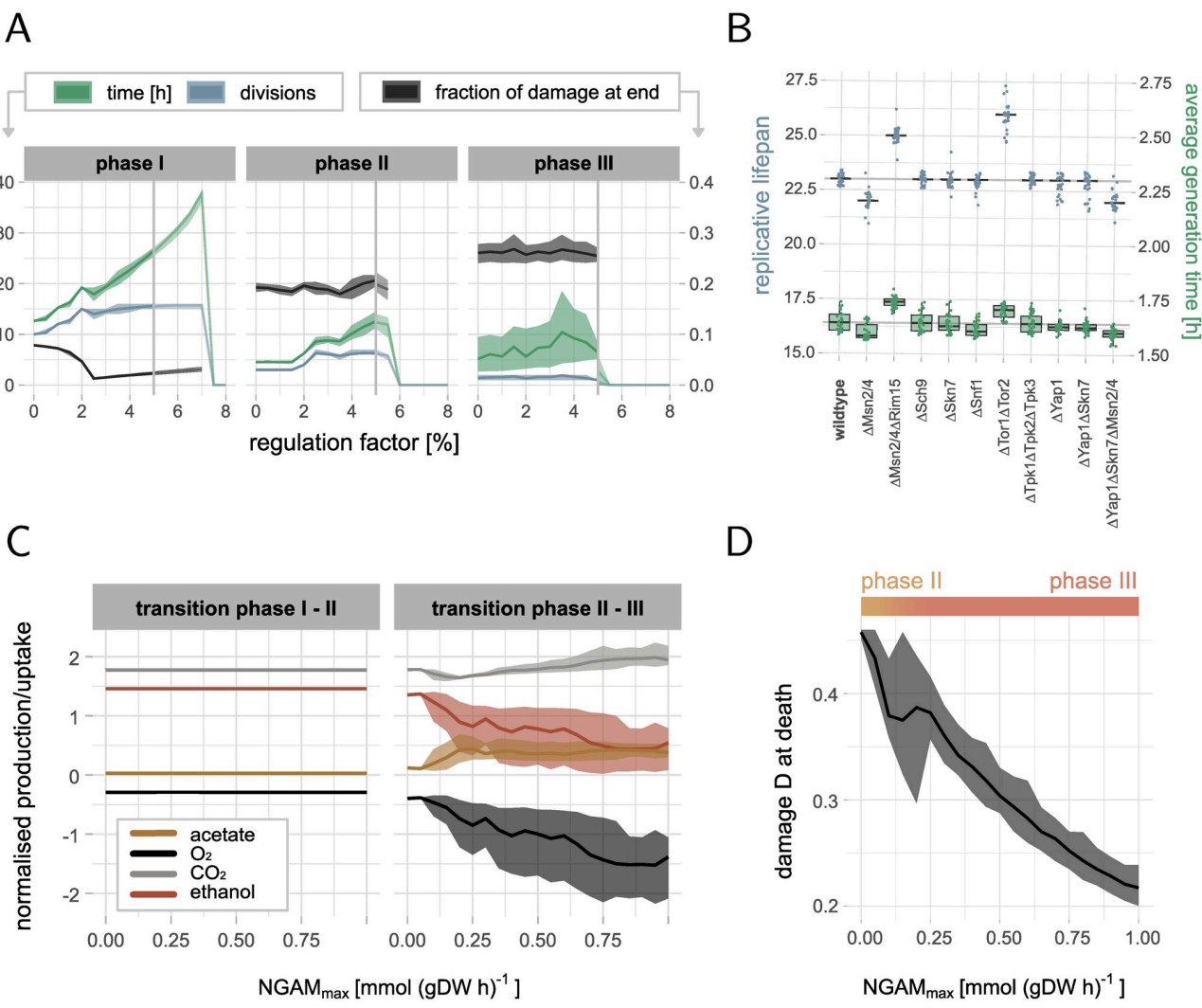

**Fig 3. Effect of regulation and NGAM on lifespan.** All distributions are based on 29 wildtype parameter sets with $f_0 \leq 5 \cdot 10^{-4}$ and $r_0 \leq 2 \cdot 10^{-2}$ that lead to 23 divisions (from data in Fig 2A). (A) Effect of the regulation factor $\epsilon$, i.e. how strong gene expression changes caused by stress signalling affect the metabolic model, on the lifespan simulations (line: mean, ribbons: 5% and 95% quantiles). Our model can only handle $\epsilon \leq 5\%$. Weak regulation $\epsilon \leq$ 2.5% mostly affects phase I, and stronger regulation $\epsilon > 2.5\%$ phase II. (B) Distributions of replicative lifespans and average generation times for cells with knockouts of signalling proteins in the different pathways of the Boolean model (line: median, box: IQR, whiskers: median ±1.5 IQR). (C) Effect of the age-dependent non-growth associated maintenance NGAM (Eq (5)) on the transition between the phases (line: mean, ribbons: 5% and 95% quantiles). Increasing cost for non-growth associated maintenance, such as damage repair, can explain the switch from fermentation to respiration in phase II, indicated by higher $O_2$ uptake, lower ethanol and higher $CO_2$ and acetate production. The fluxes are normalised by the glucose uptake rate, negative fluxes are uptake and positive production rates. (D) Damage at cell death depending on NGAM (line: mean, ribbons: 5% and 95% quantiles). Increasing NGAM leads to lower damage tolerance before cell death, that happens in phase II for low NGAM and in phase III for higher NGAM.

only because of the regulation, observed in a drop in time spent in respective phases (Fig 3A). The higher the regulation factor, the earlier the drop occurs. We therefore restricted all following analysis to $\epsilon = 0.04$.

To further understand the impact of regulation in our model, we performed knockout experiments of key proteins in the signalling pathways Snf1, PKA, TOR, Yap1 and Sln1, and combinations that are know from literature to modulate lifespan (Fig 3B). The model qualitatively predicts a lifespan increase for deletion of Msn2/4 and Rim15 while deletion of Msn2/4 alone decreases the lifespan [81]. Deletion of the TOR pathway by inhibiting the TOR complex

increases the lifespan [82]. Our model can however not predict lifespan extension by Sch9 deletion [82], being an important cross-talk protein in the included pathways. While the oxidative stress response proteins Yap1, Skn7 and Msn2/4 cannot modulate lifespan if deleted alone, only the triple deletion reduces the lifespan, indicating a robustness of the cellular response to oxidative stress. Other tested deletions, including the knockout of the PKA (corresponds to ΔTpk1–3) or the Snf1 pathway, do not have an influence on the lifespan in our simulations, in contrast to experimental evidences [82, 83], but potentially change the growth behaviour and consequently the cells' average generation times in our model, emphasising the limits of a Boolean representation of signalling and the uncertainties in the transcriptional layer.

Overall, the knockout experiments showed that the increase in the lifespan is in all cases due to a larger number of divisions in phase I, while a reduction in the lifespan is often caused by fewer divisions in phase II compared to the wildtype (S4 Fig).

## An increased ATP demand for damage repair during ageing can explain the switch from fermentation to respiration

Our model emphasised that the maximal growth phase I is particularly important for the replicative life of a cell. However, a cell cannot maintain that state forever since the pool of functional proteins decreases over time, eventually leading to a drop in the growth rate and an increase in the protein damage. To understand why cells at that point start switching from fermentation to respiration, we asked if it could be explained by an increased energy (ATP) demand while ageing, i.e. a non-growth associated maintenance (NGAM). We considered protein damage repair as one crucial type of NGAM, and therefore modelled $NGAM(t)$ to be linearly dependent on the fraction of damaged components $D(t)$ in the cell (Eq (5)), reaching its maximal value $NGAM_{max}$ when the cell has a fully damaged proteome. We then simulated wildtype cells with increasing $NGAM_{max}$ to evaluate the effect of the added ATP cost on the metabolic phases.

Our model demonstrated that the growth rate drops regardless of the additional ATP cost and cells enter phase II and III. However, without NGAM ($NGAM_{max = 0}$), cells only ferment at lower rates, and do not switch to a predominantly respiratory energy metabolism (Fig 3C). Only for larger $NGAM_{max}$, cells make increasingly use of $O_2$ and produce $CO_2$, indiciating respiratory activity.

Moreover, we observed that NGAM affects the damage tolerance of the cells, measured by the fraction of damage in old cells at cell death (Fig 3D). It can be explained by the higher energy demand associated to NGAM, which the cells can at some point during ageing not satisfy anymore given the metabolic network and the resources. As a consequence, cell death is induced earlier. Only for NGAM = 0, the *in silico* cells can be solved until it has a fully damaged proteome (here $\sim 0.46\ g(gDW)^{-1}$ [84]), being however biologically unreasonable. The variability in the damage at cell death between the tested wildtype parameters is increased for low NGAM (NGAM$< 0.25\ [mmol(gDWh)^{-1}]$). In this regime, cells do not yet manage to enter phase III, and the time between the last division and death can vary a lot depending on the underlying damage formation and repair, explaining the observed non-monotonic behaviour of the measured mean fraction of damage at cell death.

In accordance with the value used in the yeast consensus model [29], we picked $NGAM_{max}$ = 0.7 $[mmol(gDWh)^{-1}]$ in other simulations, leading to a damage tolerance of about half of the proteome in our model.

## Lifespan can be modulated by intervention in specific metabolic phases

Next, we asked if it is possible to control lifespan in our model by enhancing or repressing the right processes in the right moment. We performed deletions and overexpressions of all enzymes, as well as of combinations of isoenzymes that catalyse the same reaction, and analysed the resulting replicative lifespans compared to the wildtype (Fig 4A and 4B, S1 File). All simulations were performed for the respective perturbation during the whole life, only in

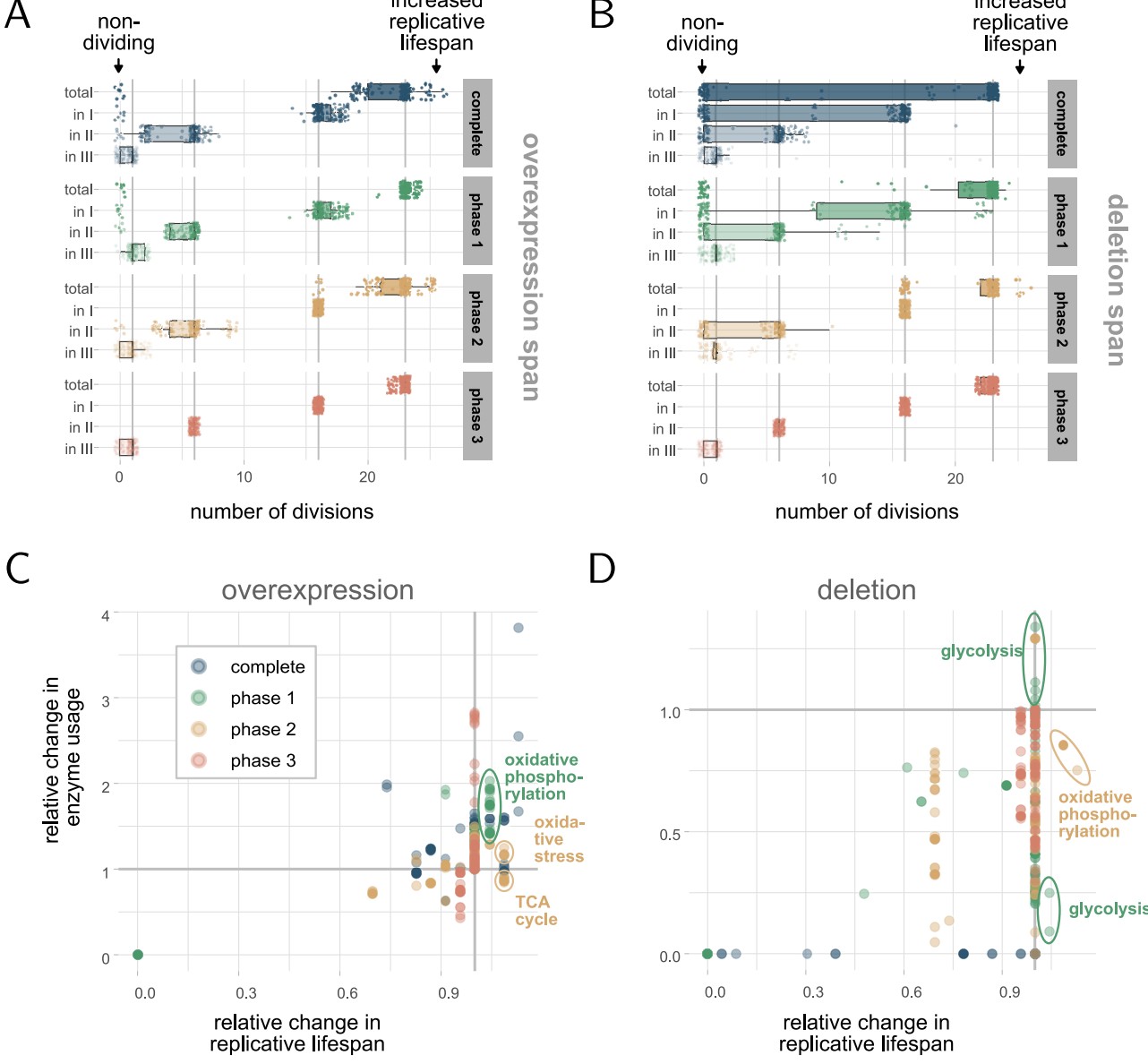

**Fig 4. Effect of enzyme perturbations on lifespan.** The simulations are based on $f_0 = 0.0001$ and $r_0 = 0.0005$ (as in Fig 2B) and perturbations (deletion or overexpression) in individual enzymes or isoenzyme combinations (140 + 23 cases) for different phases in the metabolic model. (A-B) Distribution (line: median, box: IQR, whiskers: median ±1.5 IQR) of the number of divisions in total and in particular phases upon overexpression or deletion of enzymes during specific times (facets) in relation to the wildtype (grey lines). An intervention in a specific phase can have a different effect than an intervention over the whole lifespan. (C-D) Relation between replicative lifespan and total enzyme usage relative to the wildtype. Each dot represents one simulation with the perturbation in a specific phase of an enzyme/isoenzymes. Both enzyme deletion and overexpression can lead to an increased but also decreased total usage of the enzyme compared to the wildtype.

phase I, only in phase II and only in phase III. Deletion in the model corresponds to restricting the usage of the enzyme(s) to 0. In contrast, overexpression is modelled by constraining the usage of the enzyme(s) to 150% of the optimal usage after the regulation step.

## Enzyme perturbations can enforce metabolic adaptions during replicative ageing

To gain further insight on how those enzyme perturbations affect the replicative lifespan, we studied the changes in the cumulative enzyme usage compared to the wildtype. We further looked for patterns between enzyme usages, related pathways and effects on the number of divisions (Fig 4C and 4D and S5 Fig). The data that the following results are based on can be found in detail in S1 File.

A perturbation of an enzyme can naturally result in an increase or decrease in the enzyme usage compared to the wildtype. Here, we showed that both can have a similar effect on the replicative lifespan of the cell. An increase in the lifespan can be induced by an increased usage of an enzyme, likely by enhancing processes that are beneficial for the number of divisions. On the other hand, a similar increase in the lifespan can arise from using less of a certain enzyme, indicating that high usage of this enzyme is disruptive for certain processes that correlate with lifespan. Interestingly, we found cases where an overexpression leads to a decreased usage (Fig 4C, marked with TCA cycle) and a deletion can result in an increased usage (Fig 4D, marked with glycolysis) of the enzyme in relation to the wildtype, showing that cells compensate for the loss or overuse in the successive metabolic phases, caused by altered preconditions.

To study more closely how the cells adapt to the perturbations (S1 File), several scenarios have been tested. We found that prolonging the time spent in phase I, can result in an increase of cell divisions. A prolongation of phase I can be reached by overexpressing particular enzymes from the oxidative phosphorylation pathway in phase I, or by deleting particular enzymes in the glycolysis in phase I. More specifically, we found that those enzymes typically shorten lifespan instead when overexpressed over the whole lifespan (S5A Fig, e.g. NDI1, TIM11, OLI1, several ATPs, and S5B Fig, e.g. PGI1, TPI1). Overexpression of NDI1 has been associated with ROS accumulation and apoptosis like cell death [85]. It is possible that the increased ROS production is beneficial in phase I whereas if present during the whole lifespan becomes deleterious. One can speculate that the deletion of glycolytic enzymes is likely to force flux through the glycerol pathway in the case of the TPI1 deletion or the pentose phosphate pathway for the PGI1 deletion. This probably slows down the overall growth rate and gives more time for protein repair. When enzyme capacity and the energy demand for repair starts to become limiting the added protein burden of the forced flux becomes deleterious.

Another possibility to increase lifespan is enhancing certain enzymes that remove ROS/ RNS, being beneficial in both phase I and II. Similarly, overexpressing particular enzymes in the TCA cycle can as well be advantageous in those two phases. Effectively, all cases described so far lead to a decreased growth rate that is responsible for an increase in lifespan. It is a consequence of forcing a bit of respiration already in phase I. In addition, supporting a faster switch from phase II to III can lead to more cell divisions, for instance by deleting specific enzymes in the oxidative phosphorylation in phase II. In those cases, phase III is prolonged instead. Also here, the respective enzymes are typically only beneficial for the lifespan when deleted in phase II but not over the whole lifespan (S5B Fig, e.g. COX12, COX13).

## Enzymes responsible for removing hydrogen peroxide affect lifespan

Lastly, we investigated in more detail how sensitive the lifespan is to changes in the enzymes added for ROS/RNS production and removal (S5B Fig). We found that the system is in general

robust towards perturbations in those enzymes. Most perturbations do not affect the lifespan, with few exceptions [86]. Double deletion of Trx1 and Trx2 or the deletion of Trr1, enzymes that involved in the transformation from $H_2O_2$ to water, are harmful for cell growth and divisions in all three phases [87]. Double deletion of Sod1 and Sod2, enzymes that create hydrogen peroxide of superoxide, show similar behaviour [86]. Further, we observed that overexpression of Sod1, Sod1 and Sod2, and Gpx3 completely or in phase II reduce the replicative lifespan of the cell [81]. In those cases, cells transition faster to phase III, potentially mediated by the enhanced respiratory activity. In contrast, overexpression of Trx2, Trx1 and Trx2 and Trr1 in the same phases increase lifespan, likely by enhancing removal of reactive oxidative species more efficiently in phase II but also by prolonging phase III.

## 3 Discussion

Here, we presented a novel multi-scale model consisting of the metabolism, stress signalling and damage accumulation in the budding yeast *S. cerevisiae* that allowed us to study key features of replicative ageing. We incorporated reactive oxygen and nitrogen species (ROS and RNS), as a crucial interface between the metabolism and ageing, and accounted for the asymmetric damage segregation at cell division. The model is based on established modelling techniques in Systems Biology, such as Boolean modelling, flux balance analysis and ordinary differential equations, but stands out due to the combination of the three modules to a larger interconnected model (Table 1), that tackles the challenge to deal with the complexity and multi-scale nature of ageing. Our model could reproduce realistic values for both the replicative lifespans and generation times of yeast wildtype cells, as characterised by experiments. We further showed that a regulatory layer is crucial for replicating wildtype cells in our model. Previously proposed metabolic phases [23, 24] are a direct outcome of our model, and here we demonstrated that those phases are tightly linked to the replicative lifespan using enzyme overexpressions and deletions.

We identified the non-growth associated maintenance (NGAM) as a key feature of the metabolic phases. Traditionally, NGAM is defined as the substrate yield that is used for other processes than growth [88–91]. NGAM is a dynamic variable, highly dependent on for instance the metabolic state of the cell or the nutrient composition in the media. However, there is a lack of consensus in what is included in the NGAM, since there are no direct ways to experimentally assess and quantify ATP demands specific to certain processes. Here, we assumed that protein repair and replacement of damaged proteins with functioning proteins are the main contributions to the NGAM, such that the NGAM should scale with protein damage. In the model, NGAM is defined to be linearly dependent on the damaged protein fraction, which increases with age. This simplified age-dependent definition provides a new perspective on the cost, based on the idea that the more damage there is in a cell, the more energy it needs for repair and degradation.

We tested how sensitive the model is to the maximal value of NGAM, when damage levels are the highest, and studied its effect on ageing phenotypes. We found that the increased ATP cost connected to the NGAM has a major effect when lower enzyme availability causes a decrease in the growth rate, since in that moment damage accumulation starts to increase in speed. More specifically, the NGAM changes the dynamics of the switch from fermentation to respiration happening in phase II. Leupold et al. [23] speculated that an increase in cellular volume and thus a decreased volumetric substrate intake lead to the switch of metabolic phenotype by inhibiting the carbon uptake rate [92]. In our model, NGAM plays a crucial role in determining the ratio between respiration and fermentation in the mixed metabolism that characterises phase II, and the model showed that without NGAM the switch is not induced

properly, but fermentation is mainly slowed down. Since cellular respiration has a higher ATP yield compared to fermentation, we propose that an increased ATP demand due to a non-zero NGAM together with a decreasing capacity to take up nutrients is another explanation for the metabolic switch during replicative ageing.

Gene regulation is a crucial mechanism to adapt to stressful conditions and to ensure that the right proteins are expressed in the proper time. In our model, regulation acts upon internal stress caused by ageing. It helps to increase the replicative lifespan, even under nutrient rich conditions without artificial stress, by inhibiting damage formation predominantly in phase I, but also in phase II. Delaying the onset of protein damage accumulation further prolongs the cell's health span, and in that way has a positive effect on the progeny and thus the whole cell population [47]. The model demonstrated that regulatory constraints are an important extension of FBA models in the context of ageing, as they are key in predicting wildtype lifespans.

Regulation and stress have been extensively studied in relation to replicative ageing [52, 93, 94]. Gene modifications of proteins in the nutrient and stress signalling systems, such as Msn2/4 and Tor1/2, have been connected to longevity [52], and our model was able to qualitatively predict long- or short-lived mutants for some gene knockouts. Similarly, we simulated perturbations in the enzymes contained metabolic model, and were able to confirm some qualitative correlations between enzyme deletion or overexpression and lifespan, focusing on enzymes connected to ROS/RNS. Nonetheless, the model cannot capture all known relations. There are both technical reasons and knowledge gaps that could cause the discrepancies when comparing the model to an observed phenotype. Firstly, connecting transcription factor activity to changes in gene expression is a non-trivial problem. Advanced methods utilising high-throughput data can provide means to improve the connection between signalling and the metabolism, estimating probabilistic mappings between transcription factor activities and gene expression, and translating those to the metabolic fluxes [43, 95]. However, those models are non-mechanistic and highly context-dependent, and the ability to extrapolate to other conditions, where data is limited, is questionable. Secondly, the topology of the signalling network is not completely elucidated and there are still conditions under which we cannot explain responses given our current knowledge [27, 42]. Moreover, by implementing signalling as a Boolean model, we reduced the complexity of the system which automatically limits the complexity of the model responses. Lastly, even though FBA models are good in predicting exchange fluxes and qualitative changes in pathway fluxes, individual enzyme predictions remain a challenge [37].

Motivated by the distinct metabolic phases a cell undergoes during ageing, we propose an intervention span for lifespan control. Our model showed that an enzyme perturbation in a specific phase can prolong lifespan, while the same perturbation over the whole lifespan can shorten lifespan. The same thought can be reversed, and such models can help to identify enzymes that shorten lifespan when perturbed in a specific metabolic phase, but do not affect or even prolong lifespan when perturbed over the whole life. While we focused solely on lifespan extensions in the context of ageing, phase-dependent interventions using our modelling approach can have further applications. Phase I is dominated by fermentation, and genetic modifications that prolong this phase can be of great interest in industrial applications to increase production yields. For this purpose, replacing the reconstruction of the central carbon metabolism by a reconstruction of the yeast consensus metabolic network [29] together with the addition of relevant production pathways can extend the applicability of our model and enables testing of a greater variety of interventions.

Practically, to realise such a phase-specific intervention, induceable and conditionally expressed genes in genetically modified production strains have established experimental methods.

Mathematical modelling typically is a balance between biological realism and mathematical simplicity, such that also our model is naturally based on numerous assumptions and simplifications. Yet, the model we constructed constitutes a first attempt to shed light on replicative ageing from a multi-scale perspective, incorporating several hallmarks of ageing. We could replicate important features of replicative ageing, and moved a step further in understanding and utilising the connection between the metabolism and ageing. Moreover, the modularity of our approach facilitates developing and extending the model further, and translating it to other organisms. Multi-scale mathematical models, like the one presented here, are an important aid to bridge the gap between biological realism, the knowledge we have and experimental feasibility, and to test hypotheses in a complex phenomenon like ageing.

## 4 Methods

### Extension of a regulated enzyme-constrained metabolic model by production of reactive oxygen and nitrogen species and the cellular response to them

We based our work on a previously published hybrid model of nutrient signalling and the metabolic network of the central carbon metabolism, that consists of a Boolean modelling approach of the nutrient signalling pathways TOR, PKA and Snf1 combined with an enzyme-constrained flux balance analysis approach (ecFBA) via a transcriptional layer [37]. In this model, the first step is to optimise the ecFBA model for a given objective. The optimal glucose uptake flux then determines the state of glucose in the Boolean model, i.e. glucose is present if the glucose intake exceeds a critical threshold $glc_c^{in}$. A switch of the state induces a cascade of events in the Boolean model and eventually its steady state gives rise to which of the transcription factors in the pathways are active. For each transcription factor in the Boolean model the database Yeastract [49] can tell which genes are subsequently expressed or inhibited. Since also enzymes of the ecFBA model are included in the target lists from Yeastract the constraints on their usages in the ecFBA model can be altered accordingly.

In particular, for each enzyme $i$ a rank based on the number of transcription factors that up- or down-regulate it determines if the netto regulation is positive or negative. The bounds of the enzymes $e_{i,min}$ and $e_{i,max}$ in the ecFBA are then constrained according to:

$$\begin{aligned} \textbf{upregulation}: \quad & e_{i,min} \leftarrow e_{i,min} + \Delta_i \cdot \epsilon \\ \textbf{downregulation}: \quad & e_{i,max} \leftarrow e_{i,max} - \Delta_i \cdot \epsilon, \end{aligned} \quad (1)$$

with $\epsilon$ being a regulation factor, and $\Delta_i$ being the range of enzyme usages that the model can take (result from enzyme variability analysis) without varying the objective value of the original optimisation up to the flexibility $\gamma$. This is necessary to ensure that the regulation does not induce the infeasibility of the model and that the network has enough flexibility to reallocate the resources caused by the regulation. The solution of the ecFBA with the new constraints corresponds to the regulated metabolic network.

We used the same methodology but increased the size of the metabolic network by including reactions in the ecFBA that produce and remove reactive oxygen (ROS) and nitrogen (RNS) species (Fig 1A). The major source are electrons that escape from the electron transport chain in the mitochondria during cellular respiration [17] and react with oxygen to produce superoxide. Superoxide can via RNS be transformed to hydroxyl radicals which can oxidise and thus damage proteins. A second way to create protein damage from superoxide is via hydrogen peroxide. Hydrogen peroxide in the model is either removed or reacts further to become the dangerous hydroxyl radical [14, 18, 21, 22, 50–52]. Subsequently, certain levels of

oxidative stress in the cell trigger stress signalling [21]. Therefore, we added the oxidative stress sensing pathways Yap1 and Sln1 to the Boolean model (Fig 1B), that induce regulation of the metabolic network via gene regulation by the transcription factors Yap1 and Skn7 [53–67]. To improve the transcriptional layer, we extended the data from Yeastract by data from [68], that particularly focused on the effects of Yap1 and Skn7. The presence of oxidative stress in the Boolean model is steered by the production of proteins with oxidative damage in the ecFBA model, that if larger than $d^c$ switches the presence of $H_2O_2$ to 1. Similarly, the enzyme usage of Trx1/2, proteins known to regulate the Yap1 pathway as well as to activate Msn2/4, determines if the protein is present in the Boolean model, in particular if it exceeds a critical threshold $trc^c$.

In total, we added 9 new components and 13 new rules to the existing model of nutrient signalling to account for oxidative stress signalling by the Yap1 and the Sln1 pathway. Moreover, the ecFBA model was extended by 53 new fluxes, 28 new components and 13 new enzymes. For the metabolic- and signalling modules, the number of components, reactions, and rules were chosen inclusively to reflect our current knowledge of the pathways in the context of nutrient signalling and ROS/NOS, whereas the pathways added were chosen based on biological relevance. The dynamical module of cellular growth and damage accumulation is constructed as a minimal model.

## Multi-scale model construction of the regulated cellular metabolism and replicative ageing

We used the extended regulated enzyme-constrained FBA model described above and optimised it for maximal growth and parsimony. Besides the resulting optimal value of the growth rate $g(t)$, defined by the biomass equation, the model now also outputs a protein damage formation rate $f_m(t)$ that is caused by oxidative stress. $f_m(t)$ is defined by the exchange flux from protein damage (Fig 1B).

To evaluate the protein damage formation over time, we incorporated a third module: a dynamical model based on a simple system of ordinary differential equations. The states are the cell's dry weight $M(t)$, its fractional intact protein content $P(t)$ $[g(gDW)^{-1}]$ and its fractional damaged protein content $D(t)$ $[g(gDW)^{-1}]$. The latter two can be transformed between each other, however the total fraction of proteins $P(t) + D(t)$ is assumed to remain constant over time ($\Leftrightarrow \frac{d}{dt}(P(t) + D(t)) = 0$). This assumption is based on experimental data measuring the relative change in protein abundance of a set of proteins over replicative ageing [69] multiplied by absolute proteomics data [70], exhibiting only minor changes ($< 5\%$) in the estimated total abundance during ageing.

$$\frac{dM(t)}{dt} = g(t)M(t) \tag{2}$$

$$\frac{dP(t)}{dt} = -(f_m(t) + f_0)P(t) + r_0 D(t) \tag{3}$$

$$\frac{dD(t)}{dt} = +(f_m(t) + f_0)P(t) - r_0 D(t). \tag{4}$$

Besides the parameters that are directly obtained from the solution of the ecFBA, we included a non-metabolic damage formation rate $f_0$ to account for all other processes that produce damage, and a damage repair rate $r_0$ that represents all mechanisms that repair damaged proteins.

The solution of the ODE model (2)–(4) for a small time step determines the fraction of the enzyme pool that is available in the ecFBA model in the next time step. Moreover, an

increasing amount of damage increases the non-growth associated ATP cost (NGAM) in the ecFBA model, assuming that the cell needs to allocate more energy to repair damage. In particular,

$$NGAM(t) = \frac{D(t)}{P(t) + D(t)} \cdot NGAM_{max}. \tag{5}$$

In this way, we can simulate ageing as the accumulation of damage in the cell over time. Consequently, the amount of enzymes available in the ecFBA model shrinks, and the metabolic fluxes are forced to adapt in the course of the cell's lifespan.

As soon as a cell has built up enough in biomass, $M(t_d) = s^{-1}M(0)$, it divides into a mother and a daughter cell, according to a size proportion $s \in [0.5, 1]$ that corresponds to the fraction of biomass that remains in the mother cell at cell division. While the total fraction of proteins is constant in both cell compartments, the composition of functional and damaged proteins is determined by damage retention. The larger the retention factor $re \in [0, 1]$, the more damage is retained in the mother cell. The states of mother and daughter cell are updated according to

$$
\begin{array}{ll}
\textbf{mother} & \textbf{daughter} \\
M \leftarrow sM(t_d) = M(0) & M \leftarrow (1-s)M(t_d) = (1-s)s^{-1}M(0) \\
P \leftarrow (1-re)P(t_d) & P \leftarrow (1+re)P(t_d) \\
D \leftarrow (1+re)D(t_d) & D \leftarrow (1-re)D(t_d).
\end{array} \tag{6}
$$

Cell death occurs when the enzyme-constrained FBA becomes infeasible, i.e. when the cell is not able to obtain enough energy for maintenance and growth anymore.

By including the dynamic module for damage accumulation, we incorporated the notion of time to the regulated metabolic model, under the assumption that the steady state of the metabolism is reached fast in the time scale of replicative ageing. It further led us to introduce a time delay between the moment the cell receives a stress signal and the moment the metabolic network is affected by altered gene regulation.

A schematic view of the complete model is shown in Fig 1. All details about the individual models and the extensions made in this work for both the signalling and the metabolic reactions can be found in S1 Text, and on github https://github.com/cvijoviclab/IntegratedModelMetabolismAgeing under ModelFiles.

## Simulation details

All simulations and their analysis were performed in the programming language Julia version 1.6 [71] and were run on a normal computer with 2.3 GHz Dual-Core Intel Core i5 and 8GB RAM. The linear program (ecFBA) was solved using the JuMP and Gurobi packages. The developed model can be downloaded from https://github.com/cvijoviclab/IntegratedModelMetabolismAgeing. Model parameters, constraints and pseudo code for a lifespan simulation can be found in S2 Text.

## Supporting information

**S1 File. Collection of all simulation data of enzyme deletions and overexpressions and their effect on the respective phases.**
(XLSX)

**S1 Text. Model details.**
(PDF)

**S2 Text. Computational guidelines.**
(PDF)

**S1 Fig. Validation of Boolean model extension.** Boolean model states when turning on ($0 \rightarrow 1$) and off ($1 \rightarrow 0$) hydrogen peroxide ($H_2O_2$) depending on glucose and nitrogen availability and the presence of thioredoxins (Trx1/2). $H_2O_2$ eventually triggers active transcription factors Skn7 and Yap1, the latter only if Trx1/2 is not active. Active Trx1/2 inhibits Yap1 activation and activates Msn2/4.
(EPS)

**S2 Fig. Validation of the ecFBA model extension.** Simulation of the chemostat experiment [73] using the extended regulated enzyme-constrained metabolic model. The left panel shows exchange fluxes in the model (solid lines) compared to the experimental data (dots). The right panel shows damage-related fluxes in the model, where chemostat data is not available.
(EPS)

**S3 Fig. Complementary simulations of yeast cells.** For varying damage repair $r_0$ and non-metabolic damage formation rates $f_0$. (A) Correlation between the time cells spend in phase I and their replicative lifespan. (B) Average generation times. (C) Fraction of damaged protein content at the end of phase I. (D) Replicative lifespans. (A)-(C) are coming from the same data set as Fig 2C. The grid in (D) is further zoomed out. If there is no colour it means that the cell did not stop dividing in the simulation time, since repair is so efficient that it overcomes damage formation and retention.
(EPS)

**S4 Fig. Effect of signalling protein knockouts on phases.** Split of the number of divisions over the metabolic phases for knockouts of signalling proteins in the different pathways of the Boolean model for the cells in Fig 3B. The distributions are based on 29 wildtype parameter sets with $f_0 \leq 5 \cdot 10^{-4}$ and $r_0 \leq 2 \cdot 10^{-2}$ that lead to 23 divisions (from data in Fig 2A). A lifespan increase is mostly achieved by prolonging phase I.
(EPS)

**S5 Fig. Effect of enzymes perturbations on phases.** Absolute changes in the number of divisions in relation to a wildtype cell (23 divisions) for overexpressions (A) or deletions (B) of single enzymes or isoenzyme combinations (140 + 23 cases) in different phases in the metabolic model. The simulations are based on $f_0 = 0.0001$ and $r_0 = 0.0005$ (as in Figs 2B and 4). Enzymes that do not lead to any differences are excluded in this plot. Perturbation in a specific phase can have a different effect on the lifespan than the same perturbation over the whole life.
(EPS)

## Acknowledgments

We would like to thank members of the Cvijovic lab for valuable input.

## Author Contributions

**Conceptualization:** Barbara Schnitzer, Linnea Österberg, Marija Cvijovic.

**Formal analysis:** Barbara Schnitzer, Linnea Österberg, Iro Skopa.

**Investigation:** Barbara Schnitzer, Linnea Österberg.

**Methodology:** Barbara Schnitzer, Linnea Österberg.

**Project administration:** Marija Cvijovic.

**Resources:** Marija Cvijovic.

**Supervision:** Marija Cvijovic.

**Visualization:** Barbara Schnitzer.

**Writing – original draft:** Barbara Schnitzer, Linnea Österberg, Marija Cvijovic.

**Writing – review & editing:** Barbara Schnitzer, Linnea Österberg, Marija Cvijovic.

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
