## [Decision Letter · Decision Letter 0]

19 Apr 2022

Dear Dr. Cvijovic,

Thank you very much for submitting your manuscript "Multi-scale model suggests the trade-off between protein and ATP demand as a driver of metabolic changes during yeast replicative ageing" for consideration at PLOS Computational Biology. As with all papers reviewed by the journal, your manuscript was reviewed by members of the editorial board and by several independent reviewers. The reviewers appreciated the attention to an important topic. Based on the reviews, we are likely to accept this manuscript for publication, providing that you modify the manuscript according to the review recommendations.

Sincerely,

Kiran Raosaheb Patil, Ph.D.

Deputy Editor

PLOS Computational Biology

[LINK]

Reviewer's Responses to Questions

**Comments to the Authors:**

Reviewer #1: In this manuscript, a hybrid model developed in previous work by the authors describing the central carbon metabolism and nutrient signaling in S. cerevisiae is augmented by ROS/RNS related metabolic reactions and signaling and coupled with three ordinary differential equations depicting mass balances of fractions of intact and damaged proteins and cell biomass. The authors employed the developed model to provide a nice and insightful study of replicative yeast aging and quantify the extent of protein damage in cells.

The text is rather clear and well-structured, however some details have to be explained. Specific comments follow.

Comments:

1. The authors assume that the total protein fraction is constant over the lifespan of S. cerevisiae. It seems that this assumption is strong, knowing that protein fraction variation across growth rates in S. cerevisiae is 45-55% and that some other yeasts have even more pronounced variations over different growth rates. Considering that significant metabolic changes arise over successive replication, I think that it is important that the authors discuss this assumption. Are experimental data about the total protein fraction over replicative aging available in the literature? Can the authors comment on how sensitive this work’s conclusions would be if this assumption does not hold?

2. Following up on the previous remark, I could not find in the main manuscript or in the supplementary material (I did not check the code) is it there in the model a constraint that relates the intact and damaged proteins as a weighted sum of proteins in the biomass equation?

3. Similarly, how exactly is fm(t) computed? I think that this information and information from 2. should be in the main manuscript as it will allow to readers understand the relationships between ecFBA and the ODEs better.

4. I could not find how much is ndelay?

5. Table 1 – Perhaps I missed somewhere the definition, but what are “components”? Are these the metabolites and also the enzymes? In the table, when the size of the metabolic network is reported, the authors distinguish components and enzymes. Below, it seems that for yMSA the number of components involves the number of components from signaling and metabolic network but also the number of enzymes – this is confusing. Similarly, e.g., line 399 it is said “the ecFBA model was extended by 53 new reactions and 41 new components including 13 new enzymes” – how we should understand this statement? Please clarify.

6. Figure 2 – One can observe that glucose uptake is constant throughout the generations – ROS can inhibit multiple glycolytic enzymes, so, glucose uptake rate is likely to change over the generations. Please discuss this and acknowledge in the text. In the caption: “Stronger regulation increases the replicative lifespan and wildtype cells with more than 22 divisions cannot be achieved in this resolution if the regulation factor eps < 0.4” – is here 0.4 a typo? Should not be 0.04? Otherwise, I do not understand this statement from Figure 2, please clarify. Timescale in Figure 2B is in hours, please put in the figure. Fluxes are normalized with respect to what?

7. Can you please discuss in more detail on the nonmonotonic behavior of the curve around NGAM=0.25 mmol/gDW/h?

8. Line 121, “The objective of the metabolic model is always maximal growth, but a certain flexibility is allowed” – can you be more precise/quantitative what do you mean by “a certain flexibility”?.

9. Line 377, “constrained” instead of “constraint”

Reviewer #2: I recommend this work for publication after a minor revision.

Summary

In this modeling work, the authors examine connections between replicative lifespan of budding yeast and two chemical species: Reactive Oxygen Species (ROS) and Reactive Nitrogen Species (RNS). An interesting aspect that the authors investigated is that both ROS and RNS are harmful to yeast but also are necessary and inevitable byproducts of metabolism. Specifically, the authors examined how yeast's metabolism changes as the cell's replicative age increases. To do so, the authors developed a multi-scale model that composed of a Boolean modeling of intracellular signaling, flux balance analysis (FBA) of metabolism, and a dynamical systems model of cell growth and an accumulation of damaged proteins. This model recapitulated known features of replicating aging (e.g., avg. lifespan and subsequent cell division taking longer than the previous division). The modeling also revealed new features: how three distinct metabolic phases that the authors identified affect a yeast's replicative potential. Using this result, the authors proposed ways to control a yeast's lifespan.

Major points:

1. In their yMSA, it's unclear where the number of each components and rules listed in Table 1 came from. Importantly, it's unclear to me how randomly taking out some of these components or rules would qualitatively change the authors' conclusions. Is this a minimal model? I don't expect the authors to rerun their model and repeat all the work by randomly leaving out some of the modules. But I think some justification for choosing so and so many components and rules would be helpful to the readers.

2. Why is it surprising that overexpressing Sod1, Sod2, Gpx3, and Glr1 completely or in phase II decrease the replicative lifespan? Give some explanation for this result.

3. Related to 2: several results are given without an intuitive / biological explanation of why they make sense. I know that it's difficult to do so for complex pathways but I think the readers would be better served even if some speculation is given for the results.

Reviewer #3: Building a multi-scale model of yeast replicative ageing (including an enzyme-constrained FBA model, a Boolean model of nutrient signalling pathways and dynamic model of protein damage accumulation and cell growth), Schnitzer et al aimed to investigate changes in the metabolism during replicative aging. Their model is unique due to the integration of three modules.  

The model could reproduce close-to-experimental values for average replicative lifespan and generation times of wild-type yeast cells. The model also predicts features of replicative aging with distinct metabolic phases.

This is a timely work that aims to fill in an essential gap in quantitatively understanding aging and lifespan at the single-cell level. While I am overall positive, implementation of the following suggestions would be needed as they would further improve the impact of this work.

1. While the size of the individual modules of this multi-scale model is provided in a table, it would be good to also explain/inform the reader about which parameters have been empirically known and which have not. Also, any quantitative assessment of the robustness of the overall model should ideally be made.

2. As the authors also acknowledge, the phenotypic effects of some of the gene deletions cannot be reproduced by the model. It would be good if the authors at least tried to explain these shortcomings in a gene-to-gene basis (i.e. what changes in the model would make the model predict the phenotypic effect of gene x’s absence?)

3. Protein damage repair could be one type of NGAM, but could there be others? Potential other processes should be mentioned.

4. There are previously-published yeast-aging-focused or time-dynamic single-cell-modeling-focused papers that should have been cited due to their relevance to the different sections of this manuscript:

http://doi.org/10.1016/j.celrep.2019.07.082

http://doi.org/10.1186/s12859-019-2921-3

http://doi.org/10.1186/s12918-015-0240-5

5. In the absence of direct experimental validations, the sections on NGAM and enzyme perturbations are not currently looking to be very strong. They are simulation results obtained from a model with lots of components/parameters without full in vivo values. This is not meant to be a major criticism; aging is a complex process and it is not surprising that it is taking a multiscale model to predict features to be experimentally tested. To improve the manuscript in these aspects, it would be good if the authors found as much relevant information from experimental literature as possible to validate bits and pieces of the predictions of the model. While this understandably cannot be comprehensive, it would help with the validation of the model.

**Have the authors made all data and (if applicable) computational code underlying the findings in their manuscript fully available?**

Reviewer #1: Yes

Reviewer #2: Yes

Reviewer #3: Yes

PLOS authors have the option to publish the peer review history of their article (what does this mean?). If published, this will include your full peer review and any attached files.

Reviewer #1: No

Reviewer #2: No

Reviewer #3: No

Figure Files:

Data Requirements:

Reproducibility:

References:

---

## [Decision Letter · Decision Letter 1]

31 May 2022

Dear Dr. Cvijovic,

We are pleased to inform you that your manuscript 'Multi-scale model suggests the trade-off between protein and ATP demand as a driver of metabolic changes during yeast replicative ageing' has been provisionally accepted for publication in PLOS Computational Biology.

Best regards,

Kiran Raosaheb Patil, Ph.D.

Deputy Editor

PLOS Computational Biology

Reviewer's Responses to Questions

**Comments to the Authors:**

Reviewer #1: The authors have addressed my remarks adequately.

Reviewer #2: The authors have addressed most of my comments. I recommend publication.

Reviewer #3: The authors adequately addressed my comments and suggestions. I recommend the publication of this manuscript.

**Have the authors made all data and (if applicable) computational code underlying the findings in their manuscript fully available?**

Reviewer #1: Yes

Reviewer #2: Yes

Reviewer #3: Yes

PLOS authors have the option to publish the peer review history of their article (what does this mean?). If published, this will include your full peer review and any attached files.

Reviewer #1: No

Reviewer #2: No

Reviewer #3: No

---

## [Editor Report · Acceptance letter]

27 Jun 2022

PCOMPBIOL-D-22-00380R1 

Multi-scale model suggests the trade-off between protein and ATP demand as a driver of metabolic changes during yeast replicative ageing

Dear Dr Cvijovic,

I am pleased to inform you that your manuscript has been formally accepted for publication in PLOS Computational Biology. Your manuscript is now with our production department and you will be notified of the publication date in due course.

With kind regards,

Anita Estes
